# Containment of COVID-19: Simulating the impact of different policies and testing capacities for contact tracing, testing, and isolation

**Vincenzo G. Fiore**[1]*, **Nicholas DeFelice**[2], **Benjamin S. Glicksberg**[3,4], **Ofer Perl**[1], **Anastasia Shuster**[1], **Kaustubh Kulkarni**[1], **Madeline O'Brien**[1], **M. Andrea Pisauro**[5,6], **Dongil Chung**[7], **Xiaosi Gu**[1]

**1** Department of Psychiatry, Icahn School of Medicine at Mount Sinai, New York, New York, United States of America, **2** Department of Environmental Medicine and Public Health, Icahn School of Medicine at Mount Sinai, New York, New York, United States of America, **3** Department of Genetics and Genomic Sciences, Icahn School of Medicine at Mount Sinai, New York, New York, United States of America, **4** Icahn School of Medicine at Mount Sinai, Hasso Plattner Institute for Digital Health at Mount Sinai, New York, New York, United States of America, **5** Department of Experimental Psychology, Wellcome Centre for Integrative Neuroimaging, University of Oxford, Oxford, United Kingdom, **6** Centre for Human Brain Health, University of Birmingham, School of Psychology, Birmingham, United Kingdom, **7** Department of Biomedical Engineering, Ulsan National Institute of Science and Technology, Ulsan, South Korea

* vincenzo.fiore@mssm.edu

**Data Availability Statement:** The entire codebase for these simulations is freely available as OSF repository (DOI: 10.17605/OSF.IO/QF8WC).

## Abstract

Efficient contact tracing and testing are fundamental tools to contain the transmission of SARS-CoV-2. We used multi-agent simulations to estimate the daily testing capacity required to find and isolate a number of infected agents sufficient to break the chain of transmission of SARS-CoV-2, so decreasing the risk of new waves of infections. Depending on the non-pharmaceutical mitigation policies in place, the size of secondary infection clusters allowed or the percentage of asymptomatic and paucisymptomatic (i.e., subclinical) infections, we estimated that the daily testing capacity required to contain the disease varies between 0.7 and 9.1 tests per thousand agents in the population. However, we also found that if contact tracing and testing efficacy dropped below 60% (e.g. due to false negatives or reduced tracing capability), the number of new daily infections did not always decrease and could even increase exponentially, irrespective of the testing capacity. Under these conditions, we show that population-level information about geographical distribution and travel behaviour could inform sampling policies to aid a successful containment, while avoiding concerns about government-controlled mass surveillance.

## 1. Introduction

In December of 2019, a cluster of cases of pneumonia was recorded among people associated with the Huanan Seafood Wholesale Market in Wuhan, Hubei Province in China [1]. They were infected with a novel strain of virus, the severe acute respiratory syndrome coronavirus 2

**Funding:** VGF is funded by the Mental Illness Research, Education, and Clinical Center (MIRECC VISN 2) at the James J. Peter Veterans Affairs Medical Center, Bronx, NY. NDF is supported by National Aeronautics and Space Administration [ECOSTRES18-0046], National Institute Environmental health sciences [R24ES028522, UH3OD023337]. DC is supported by the National Research Foundation of Korea [NRF-2018R1D1A1B07043582]. XG is supported by National Institute on Drug Abuse [grant numbers: R01DA043695, R21DA049243] and National Institute of Mental Health [grant number: R21MH120789, R01MH124115, R01MH122611]. The funders had no role in study design, data analysis, decision to publish, or preparation of the manuscript.

**Competing interests:** The authors have declared that no competing interests exist.

(SARS-CoV-2). In a few months, the virus spread rapidly around the world and with no vaccine or identified treatment, it forced a growing number of countries to implement robust non-pharmaceutical policies of social distancing, such as stay-at-home orders. Over the course of 2020, the World Health Organization has reported that these mitigation strategies, where applied, have been successful in containing new daily infections, at least temporarily [2]. However, these measures have come at a high social and economic cost, leading policy makers to consider different follow-up strategies for mitigation and containment [e.g. see: 3, 4]. Thus, with several vaccine candidates showing early positive results [5], the objective for the upcoming months has shifted towards allowing as many people as possible back to work or school safely, while reducing the socio-economic impact of the pandemic, and avoiding or mitigating new waves of infections that could overwhelm the healthcare system [6].

To this end, the World Health Organization has suggested that enhanced capacity for contact tracing and testing is necessary to continuously monitor the intensity and geographical spread of the virus, detect new outbreaks at their onset, isolate (i.e., quarantine) new infections, and prevent pre-symptomatic or asymptomatic transmission [7, 8]. Contact tracing, testing and isolating potential vectors of the disease constitute the key public health process that has been used for decades to break the chain of transmission of an infectious disease [9–11]. This process aims to identify the individuals who have come into contact with an infected person, promoting targeted isolation whenever necessary, to prevent new viral shedding. Unfortunately, SARS-CoV-2 is characterised by high transmissibility [12–15], as well as a viral shedding onset that precedes the manifestation of symptoms [13, 16, 17], and that occurs even in asymptomatic and paucisymptomatic infections [18–21], i.e., subclinical infections not characterised by readily observable symptoms. Given these characteristics, the process of contact tracing and testing cannot rely only on symptom-based isolation and tracing, but it should also actively target and prevent presymptomatic and asymptomatic/paucisymptomatic transmission [17, 18, 22].

Many policy makers have already vastly increased testing and contact tracing capacity [2]. However, they are also required to strike a balance between the need to trace back the movements of those individuals who have tested positive for the virus (and those of her contacts) and the concern that legitimate health-related tracking policies may deteriorate into a form of mass surveillance. This trade-off comes in a time of crisis for democracies across the world with the associated mistrust of government communication, a phenomenon that reduces voluntary compliance to tracking methods. Furthermore, despite the general consensus that "more is better", there still remains a crucial factor to be determined: the order of magnitude of the resources required to appropriately identify, test and isolate a number of pre-/a-/paucisymptomatic infections that would reduce the SARS-CoV-2 transmission to a tolerable societal risk [23]. It is also unclear how these estimates may dynamically change depending on the type of non-pharmaceutical policies in place [14, 24–26], or as a function of efficacy and reliability of testing and tracing [27, 28]. Finally, it has not been directly examined whether an increase in testing capacity, albeit necessary, would be in itself sufficient to avoid the large number of undetected infections that have been reported across multiple countries [29–32]. This uncertainty leaves policy-makers to rely on trial and error approaches which may easily lead to new upsurges of infections, as reported even in countries and regions that seemed to have successfully controlled or avoided the first wave of infections [2].

Here we address these open questions, identifying the optimal test-and-trace resources under different scenarios, by focusing particularly on presymptomatic and asymptomatic/paucisymptomatic transmission. We used a series of multi-agent simulations [33, 34] to highlight emergent dynamics in the interaction between agents, environment, viral transmission and testing policies. In the first set of simulations, we estimated the testing capacity that would

allow for the identification and isolation of a number of infections sufficient to break the chain of transmission of SARS-CoV-2. We tested the sensitivity of this measure along four dimensions, in a 3 (relative transmission efficiency) x 5 (contact tracing and testing efficacy) x 2 (percentage of asymptomatic/paucisymptomatic infections) x 3 (size of secondary infection clusters) condition design. Specifically, three different levels of disease transmission efficiency determined the increment of new daily infections (e.g., due to different non-pharmaceutical mitigation policies in place [14, 24]). Five different rates of contact tracing and testing efficacy determined the number of infected subjects that were found or missed by the testing process (e.g., due to false negatives or untraced contacts [17, 29]). Two estimates to determine the percentage of asymptomatic or paucisymptomatic infections [18, 19, 35], which were used to define the percentage of the infected population that would not self-isolate after contracting the disease. Finally, three conditions of secondary infection cluster size were used to allow each infected agent to transmit the virus up to 1, 8 or 25 healthy agents, daily, depending on the condition.

In the second set of simulations, we focused on the hypothesis that population-level analysis of geographical distribution and travel behaviour could be used to improve randomised surveillance testing methods [3, 4, 36], so aiding the contact tracing process in finding those infected agents that were left undetected.

## 2. Methods

### 2.1 Key parameters for the simulated scenarios

All results are based on simulations that started with a healthy population of 100,000 agents (each agent representing a person), distributed on a map depending on the population density of the area analysed (Fig 1). For day 1 only, each agent was randomly assigned a geographic location, and had a .05% probability of becoming infected, resulting in a randomly generated number of infected agents (49.3±7.3 across scenarios). These differences in the initial conditions led to diverging scenarios in terms of the number of infected people, active outbreaks and the difficulty of containment, approximately replicating the estimated numbers two weeks to ten days prior to establishing the lockdown measures in France, in March 2020 [37]. Before the start of the simulation, each agent was pre-assigned to one of five symptomatology categories, used only if the agent became infected. The five categories included: asymptomatic or paucisymptomatic (20% or 30%, depending on the condition), symptomatic but not requiring hospitalization (65% or 55%, depending on the condition), symptomatic and requiring hospitalization (10%), symptomatic and requiring intensive care (4%) and symptomatic and requiring intensive care, but will not survive (1%; Fig 2a)[18, 35, 38]. At present, studies and reports do not yet agree on the relative percentages of the symptomatic infections, due to the differences among regions and countries in the methods for testing and monitoring the infections in the populations and requirements for hospitalization. Therefore, the settings concerning the relative distributions of the symptomatic infections have been included to ease future developments of the codebase, while for the present simulations, we assumed that all symptomatic infections entered isolation the day they displayed symptoms. The value of 20% for the asymptomatic or paucisymptomatic infections was determined using a weighted mean between two key studies reporting the percentage of asymptomatic infections in the Town of Vo', in Italy [18] and in the cruise ship Diamond Princess [35]. The second value of 30% was considered as a plausible alternative [19], so as to explore variations in testing capacities and analyse the consistency of the effects determined by the variations in contact testing efficacy (cf. Table 1). The number of days required to develop symptoms, if any (normal distribution: $\mu = 5.5$; $\sigma = 2$; skewness = .6; Fig 2b), the number of days to reach full recovery, with a bimodal distribution

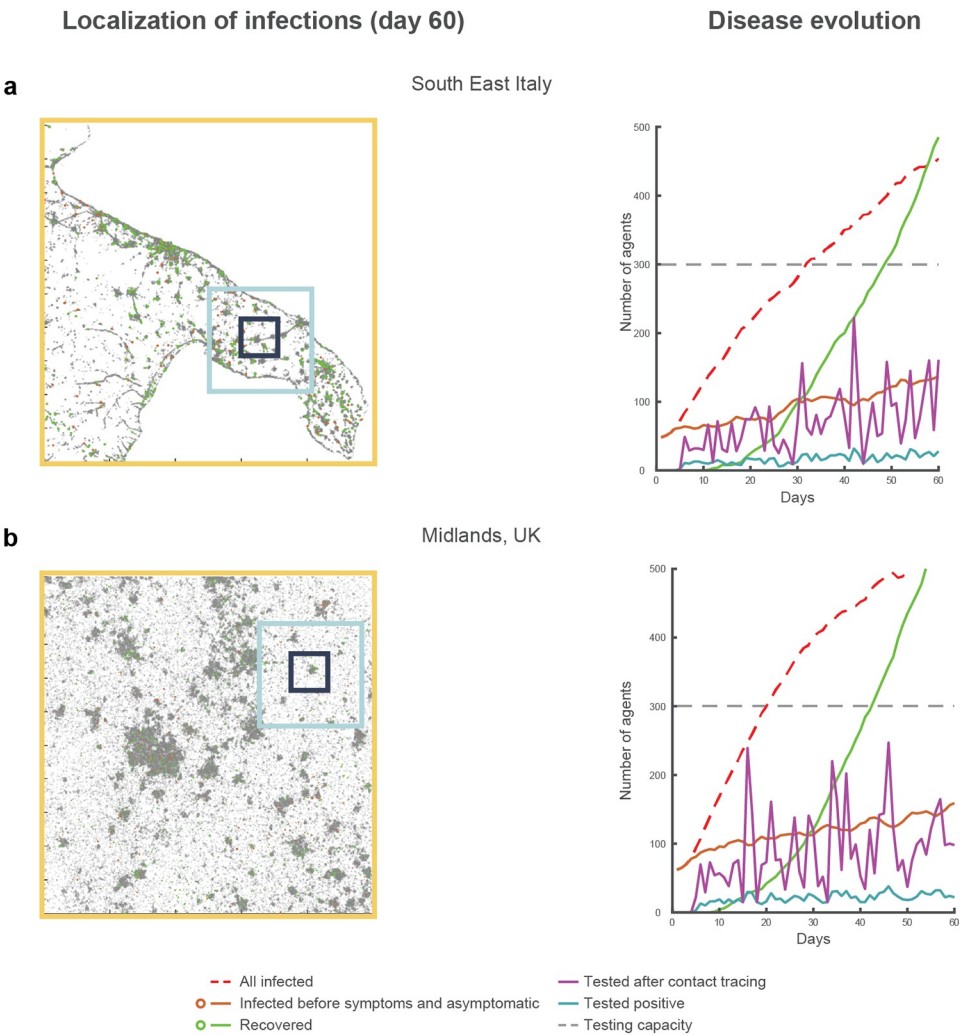

**Fig 1. Simulated evolution of the virus transmission over two regions.** Illustration of two different simulations for the scenario 1 (i.e., identical seed) for the maps of southeast Italy (a) and the Midlands in UK (b). In the raster plots on the left, the dots represent the locations of the entire population of one hundred thousand agents. All simulations display the (failed) containment of the disease transmission relying only on contact tracing, testing, and isolation, under the conditions of medium relative transmission efficiency (25% daily increase in the number of infections), 20% contact tracing and testing efficacy, 20% of asymptomatic infections, deterministic transmission, and a distribution of travel cohorts of 40%, 30% and 30% for the short, medium and long travel range (respectively illustrated as black, blue and yellow squares overlaid on the maps). The right panels illustrate the oscillatory dynamic characterising the number of tests performed (magenta continuous line). The peaks in the number of tests performed never exhaust the daily testing capacity set for the simulations (horizontal dashed grey line). However, the process of contact tracing and testing constantly loses traces of the infection due to the low efficacy, determining the various dips in terms of the daily number of tests performed and consequently an increase in pre-symptomatic and asymptomatic infections (brown continuous line). Thus, this process failure in the attempt to find and isolate a sufficient number of infected agents to suppress or contain the reproduction of the infection is found irrespective of the testing capacity.

due to the shorter time of recovery for the asymptomatic or paucisymptomatic [12, 13] (μ = 21; σ = 5; skewness = 0, for all agents, with values assigned to the subclinical cases subsequently divided by four, so to reach recovery within 8 days in ~99% of the cases; Fig 2c), and the time spent in intensive care units (μ = 5; σ = 1.5; skewness = 0; Fig 2d) were also predetermined [39–41]. To be noted that the settings controlling the ICU time or the recovery time for

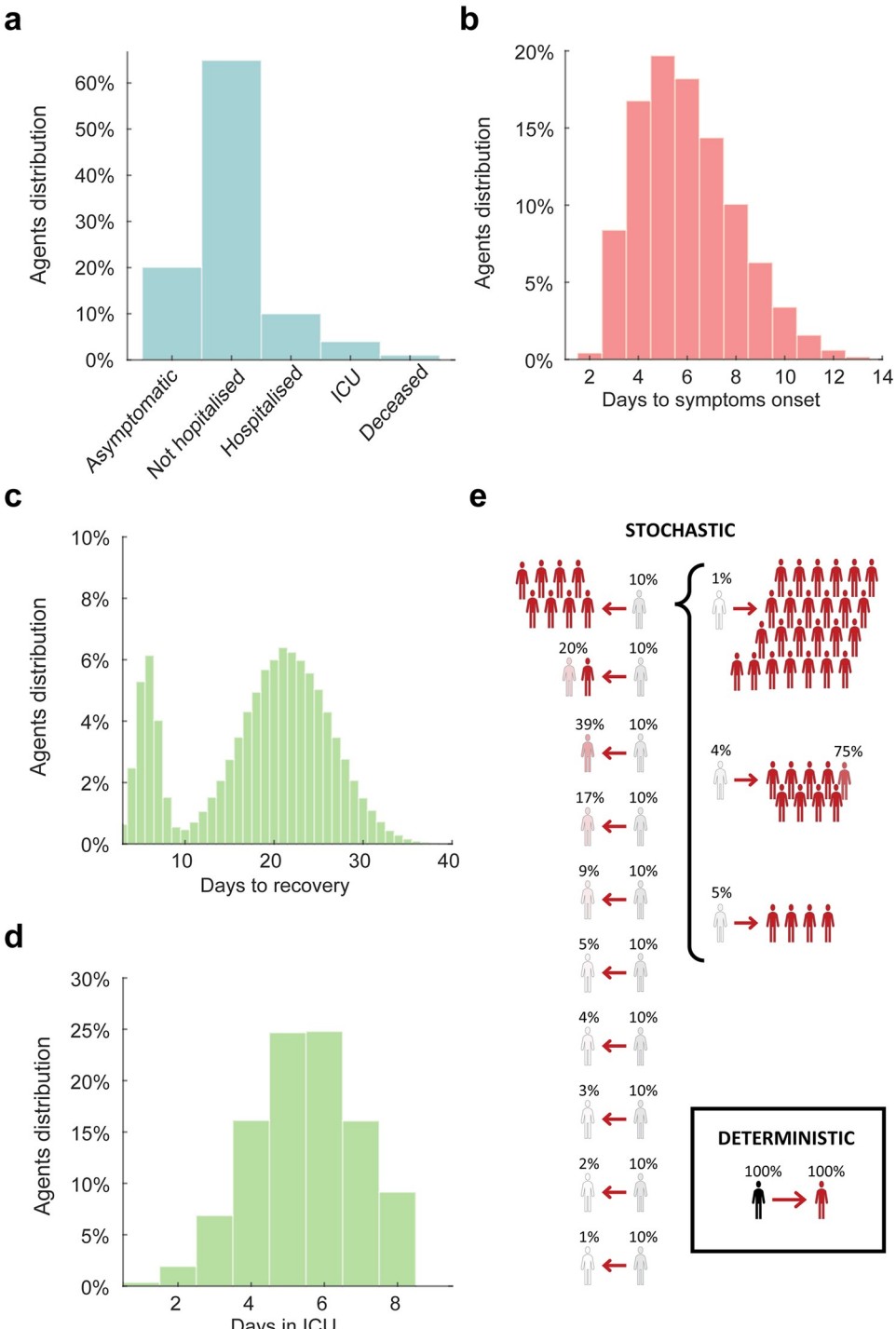

**Fig 2. Simulation settings.** The histograms represent: a) the distribution of symptoms (under the condition of 30% of asymptomatic or paucisymptomatic infections, the difference is entirely absorbed by the number of not hospitalised agents); (b) days required for the symptoms onset; (c) days required for recovery after symptoms onset; (d) days required in intensive care unit. Note that the bimodal distribution of the days to recovery is due to the presence of asymptomatic or paucisymptomatic agents who are characterised by a shorter recovery time (marking the end of viral shedding). The days spent in the intensive care unit are considered as part of the time required to recovery, when recovery is possible. Finally, the diagram in panel e) illustrates the three conditions of secondary infection cluster size used for the simulations. In the deterministic case, each agent in the population, if infected and selected for the day, transmits the infection to one agent within their travel range. In the first stochastic condition of transmission, the

population of agents is divided into 10 cohorts (each with the same assignment probability of 10%). In this case an infected agent transmits the infection to a number of targets that varies between 0 and 8 agents (e.g., a cluster size of 1.2 indicate 100% probability to reach one agent plus 20% probability to reach a second agent). In the second stochastic condition, the top cohort is split into 3 sub-cohorts with assigned probabilities of 5%, 4% and 1%. Under this condition, the number of possible targets of a single infected agent is up to 25.

symptomatic agents did not affect the results, as all symptomatic agents were isolated at symptoms onset. These settings have been included only to ease future developments of the codebase.

Finally, each agent was assigned to one of three possible "travel cohorts", defining the range of movement of the agents, and therefore their range of viral shedding: the entire map (1000x1000 pixels, comprising an entire region or metropolitan area), a medium size sector (300x300 pixels, e.g. comprising two boroughs or two separate towns, depending on the map) or a small size sector (100x100 pixels, e.g. within one city or one borough, cf. Fig 1). We simulated two conditions in terms of cohort distribution for the large, medium and small sector travel behaviour, as follows: 5%-5%-90% (*skewed* distribution) or 30%-30%-40% (*uniform* distribution). These simplified settings have been chosen only to illustrate the effects the different policies of surveillance testing have under significantly different population-wise behaviours.

For all conditions, we simulated 50 different scenarios. To allow within-scenario comparisons, we used numbered seeds controlling the random number generation. The estimations for the testing capacities were also replicated in a sub-set of 5 scenarios and populations of one million agents, keeping constant the .05% probability of becoming infected at day 1.

## 2.2 Simulation of disease transmission

To avoid overwhelmingly demanding computational resources, we did not simulate ecological behaviours for the artificial agents, as the artificial agents did not create contacts while navigating the space or form crowds (e.g., we did not simulate crowding agents in public transportation or work places). Instead, we developed a transmission mechanism that was aimed at

**Table 1. Estimated testing capacities per simulated condition.**

| | | Transmission efficiency: 15% | Transmission efficiency: 25% | Transmission efficiency: 35% |
|---|---|---|---|---|
| **Asymptomatic: 20%** | 100% contacts traced | 0.7–0.7–0.7 | 1.7–2.4–2.6 | 3.6–5.5–6.8 |
| | 80% contacts traced | 0.7–0.7–0.7 | 1.7–2.4–3.0 | 3.6–5.5–7.1 |
| | 60% contacts traced | 0.7–0.7–0.7 | 1.7–2.9–3.4 | 4.5–8.2–9.2 |
| | 40% contacts traced | 0.7–0.7–0.7 | 2.0–2.9–3.4 | 15–20–25 |
| | 20% contacts traced | 0.7–0.7–0.7 | 3.0–3.2–3.4 | 30–35–40 |
| **Asymptomatic: 30%** | 100% contacts traced | 0.8–0.8–1.0 | 2.0–3.2–3.9 | 4.0–6.8–7.9 |
| | 80% contacts traced | 0.8–0.9–1.3 | 2.0–3.7–4.0 | 4.0–6.8–9.1 |
| | 60% contacts traced | 0.8–1.0–1.3 | 2.0–3.8–6.8 | 6.0–9.8–11.9 |
| | 40% contacts traced | 0.8–1.0–1.3 | 2.3–3.8–6.8 | 35–37–40 |
| | 20% contacts traced | 0.8–1.0–1.3 | 3.3–4.0–6.8 | 50–52–55 |

Simulated testing capacities expressing the availability of tests per thousand agents, per condition. The three values in each cell represent the capacities associated with the three simulated conditions of size of secondary infection cluster (i.e., maximum number of contacts an infected agent can reach in a single simulated day: 1, 8 or 25, respectively). Background colour of an individual cell indicates reported conditions of disease containment: a white background indicates suppression of the transmission ($R_0 < 1$), a light grey background indicates containment but not suppression ($R_0 \approx 1$), and dark grey background indicates exponential growth ($R_0 > 1$). Under conditions characterised by exponential growth, the reported values indicate an approximate threshold that allows the testing capacity not to be exhausted for at least 50 days of simulation time.

prioritising the simulation of the average daily increment in the number of infections, under plausible conditions of maximum size of secondary infection clusters. To this end, the transmission of the virus was simulated starting from the daily number of infected, non-isolated agents who were within 3 days of displaying symptoms [12, 16, 42, 43], and randomly selecting a percentage of these (15%, 25% or 35%, depending on the simulated condition of relative transmission efficiency), irrespective of whether they had been already selected at any prior point in time. The three conditions of transmission efficiency simulated different reproduction numbers ($R_0$ [44]), as they determined the number of secondary infections, putatively varying as a function of the non-pharmaceutical mitigation strategies set in place [14, 24–26]. More precisely, in the absence of a testing and isolation strategy and without the isolation of symptomatic agents at symptoms onset [15], the $R_0$ associated with each simulated condition of transmission efficiency was estimated using a weighted sum that considered: 1) the probability of each infected agent to be selected each day (i.e., the relative transmission efficiency setting); and 2) the number of days an infected agent could shed the virus (cf. Fig 2c), as follows:

$$R_0 = T_e(S_t + \sum_{RH_t=1}^{8} p_{RH_t} \; RH_t)$$

Where $T_e$ stands for the transmission efficiency, $S_t$ stands for the time (expressed in days) prior to symptoms onset, $RH_t$ represents the time (expressed in days) required to reach either recovery (in mild cases), or hospitalization (in severe cases) and $p_{RH_t}$ represents the probability assigned to each $RH_t$ value. We established conservative estimates assuming infected agents were allowed to transmit the virus for up to eleven days, i.e., the three days before symptoms onset, plus up to eight days due to either recovery, or hospitalization [12, 13, 41]. These settings led to $R_0$ values of ~3.05, ~2.18, ~1.31, for the three settings of 35%, 25% and 15% transmission efficiency, respectively. These values are in keeping with current estimations for the COVID-19 [14, 41, 45–47] and allowed replicating the incremental dynamics in the number of infections reported by several countries (Fig 3).

Once the infected agents were randomly selected, with a probability determined by the transmission efficiency setting, they transmitted the infection to a number of contacts that was determined by the condition of secondary infections cluster size. We simulated one deterministic and two stochastic conditions (Fig 2e). In the deterministic condition, each day an infected agent was selected, it would transmit the virus to one single contact, randomly selected among those in the range of travel (Fig 1). Conversely, the stochastic conditions were intended to replicate the suggestion that 80% of secondary transmissions have been caused by 10% of infectious individuals [48]. To this end, all agents were pre-assigned a value at day 1, defining the number of agents that each of them could potentially infect in a single day. These values were determined using an approximation of the hyperbolic function $\left[ cluster \; size = \frac{8}{x^2} \right]$, where a vector x = [1:10] was used to generate 10 secondary infection cluster size values [8, 1.2, 0.39, 0.17, 0.09, 0.05, 0.04, 0.03, 0.02, 0.01], each assigned with equal probability (10%) to the artificial agents, resulting in a uniform distribution (Fig 2e). The integer part of the cluster size values expressed the number of contacts a single infected agent could infect in one day, whereas the decimal part expressed a probability to reach a further agent. This vector was further modified to simulate the presence of even larger secondary infection cluster sizes in a second stochastic condition. Here, we divided the top 10% cohort into three sub-cohorts, with values [25, 8.75, 4], respectively associated with probabilities of [1%, 4%, 5%]. These settings resulted in a weighted sum of 8, thus keeping the requirement of 80% secondary transmissions caused by 10% of the infectious agents [48]. For instance, on average, under the larger-cluster stochastic condition, 4 out of 100 agents were assigned a secondary infection cluster size value

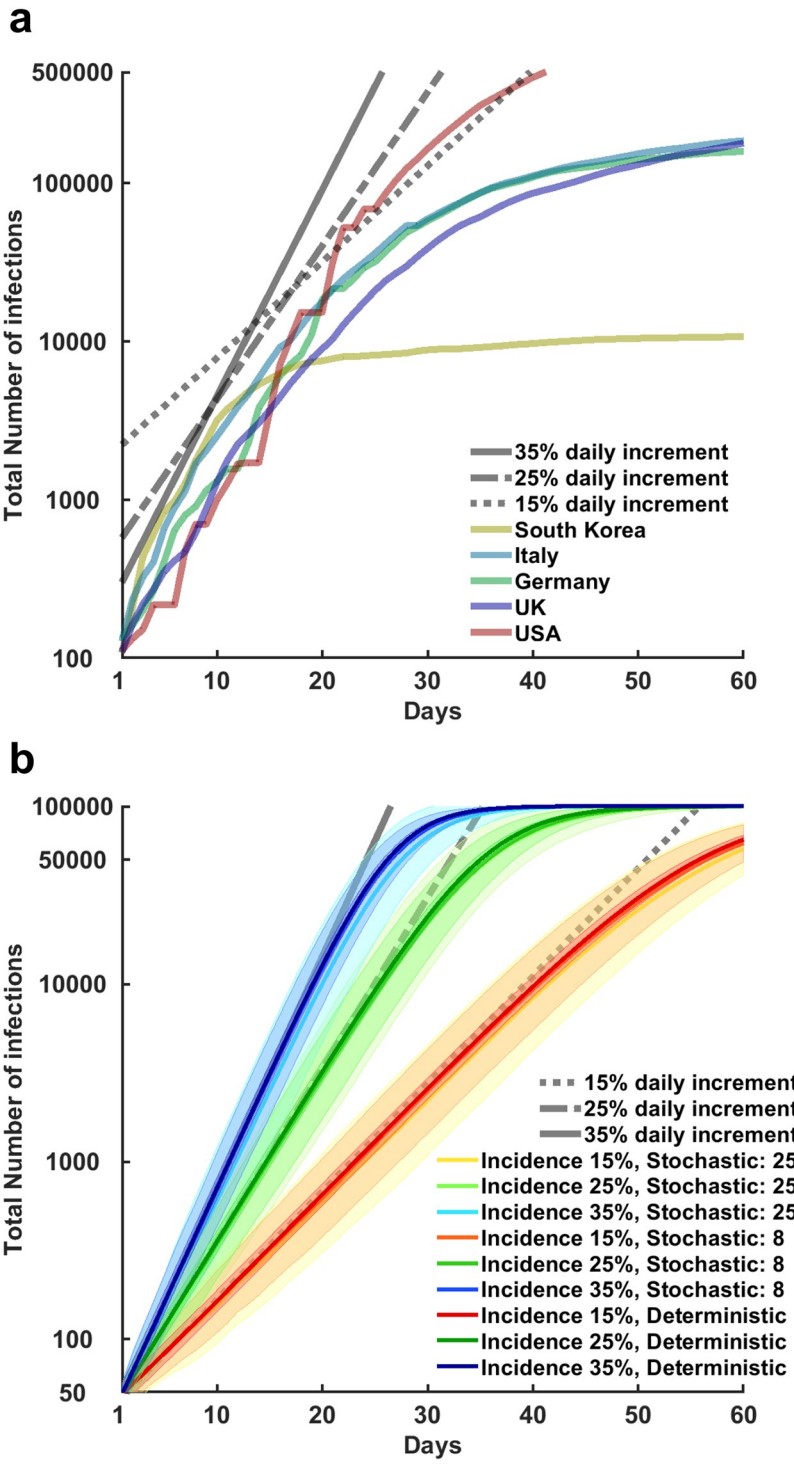

**Fig 3. Evolution in the total number of confirmed infections across multiple countries and simulated settings.** In the top panel (a), the time series illustrate on a logarithmic scale the number of total confirmed detected infections over a 60 days period for South Korea, Italy, Germany, United Kingdom, and United States (as reported by the World Health Organization). All these countries have recorded a first surge of detected infections that started in the period between late February and early March 2020. The curves have been aligned so to have at day 1 the first report of more than 100 total detected infections. The time series illustrate that the exponential growth in the number of infections was aligned initially on a 35% daily increment for all countries. After various mitigation policies became effective, the number of daily new infections (i.e. the relative transmission efficiency of the virus) followed a slower pace, at 25%

increment, 15% or less. Finally, the curve is "flattened" for most countries, within the chosen time period of 60 days. Since mitigation strategies differ across countries, the curves highlight also important differences. Notably, in South Korea the prevalence of the disease was reduced to a very small number in a few days, by relying on a rapid expansion of contact tracing and testing efforts. Conversely, European countries had to rely on stay-at-home orders to achieve the same result after failed attempt to contain the viral transmission relying only on contact tracing and testing. For the United States, the initial daily average increment of 35% is sustained for a longer time than any other country here reported, resulting in a larger number of infections within the time frame considered. In the bottom panel (b) the error bands (mean and standard deviation) illustrate on a logarithmic scale the simulated number of total infected agents in a population of 100'000 agents, varying condition of transmission efficiency and size of the secondary infection clusters. These simulated scenarios assumed no testing and isolation strategy was in place and symptomatic agents were not automatically isolated at symptoms onset. The resulting exponential increase in the number of infections replicated the dynamics described for the real case scenarios presented in panel (a). These simulations also illustrate the effect of herd immunity, as the settings in the simulations do not allow any agent to be infected twice, the $R_0$ decreases as a function of the number of infected agents that have become immune or died.

of 8.75, meaning that each day they were selected they would transmit the virus to eight other agents with 100% probability, plus a ninth agent with 75% probability. Likewise, 10 agents were assigned a cluster size value of 0.17, meaning that each day they were selected they would transmit the virus to only one agent, with 17% probability. To be noted that the average weighted size of the secondary infection clusters for the entire population was equal to 1 across all conditions, thus keeping constant the estimation for the $R_0$, across cluster size conditions (Fig 3b). Once the secondary infection cluster size was determined, the targets of the viral transmission were selected at random among the existing agents in the range of travel of the transmitter, as for the deterministic setting. Across all cluster size conditions, if a targeted agent was healthy, it was immediately infected, starting the countdown for symptom manifestation (if any). Conversely, if the contacted agent had been already infected at any point in the past, the propagation of the infection was null. Thus, with the increase in the number of recovered, actively sick or deceased agents, the probability of generating a new infection would decrease, simulating a diminished $R_0$, due to herd immunity (Fig 3b).

## 2.3 Simulation of contact tracing

Each day, the simulation of the contact tracing and testing process started after simulating the transmission of the disease. At day one of the simulated scenarios, the identity of the randomly selected .05% of the population that was infected was unknown to the system of contact tracing and testing. Thus, these agents were free to transmit the virus, starting three days before symptoms onset. However, the first day of symptoms onset, the agents that had become symptomatic were isolated and entered a pool that was updated on a day-to-day basis. This pool represented the starting point of the daily contact tracing and testing process. The agents listed in the pool were randomly selected one by one, removing them from future tracing pools and adding one positive test to the count of the day. Then, the agent under examination was used to trace the agent (if any) that had been the origin of her infection and the agents (if any) that she had infected at any prior time during the simulation. The process was repeated for all agents in the pool, until no agent remained to be examined or until exhaustion of the testing capacity (whichever came first). Those found positive via this contact tracing and testing process were isolated to prevent future transmission and entered the pool of traceable infected agents for the following day. As a consequence, all symptomatic agents eventually entered the pool of agents to be traced, by the time of symptoms onset. Conversely, asymptomatic and paucisymptomatic agents only entered the pool if found positive to a test, after being traced as the source or the target of another infected agent.

To simulate the impact of missed contacts and test sensitivity, which respectively lead to untested infections and false negative tests, five levels of contact tracing and testing efficacy

were examined (100%, 80%, 60%, 40% or 20%). This parameter determined the probability that infected agents would be found and correctly tested positive, among those that had been infected by any discovered infected agent, i.e., agents that have shown symptoms or agents that have been traced in a previous day and have tested positive, irrespective of symptom display. Importantly, a missed contact would not only result in leaving one infected agent free to transmit the disease, but it would also leave undetected the chain of transmission associated with that missed agent (i.e. the agents, if any, infected by the missed contact), until it could be traced via a new contact. This simplified mechanism conflates contacts that are missed because they were not tested at all (missed trace) with contacts that were found but then had a false negative test. In real life, a false negative test can have significantly different effects on the behaviour of a positive person in comparison with a test not performed due to a missed trace. However, in the limited settings regulating the simulated behaviour of the agents, the objective was simply to determine the number of agents that were left free to shed the virus. For instance, in a simulated scenario in which there are 100 infected contacts to trace and test, and a contact tracing and testing efficacy of 20%, 80 infected agents will avoid isolation on average. This may be due to a 25% efficacy in contact tracing jointly with 80% test sensitivity (25 contacts out of 100 are found, these 25 contacts are tested and 80% of these tests, 20, return positive), or 50% contact tracing efficacy with 40% test sensitivity (50 contacts out of 100 are found, and after being tested, 40% of the tests, 20, return positive). I.e., agents missed = traceable agents * (1- test efficacy * test sensitivity).

The simulations allowed for perfect record keeping of the actual contacts of each infections, so we implemented a system that could also simulate the number of negative tests per each positive one. This was performed dynamically to represent the different challenges in finding positive contacts, depending on the percentage of infected agents in the entire population. For each positive test found with contact tracing, the simulations added a number of negative tests, which contributed to reach the daily maximum testing capacity. This was equivalent to the number of healthy agents per active (non-isolated) infected agent, updated daily, up to a maximum of 20 negative tests per each positive one. For instance, under a scenario in which 50 thousand healthy agents and 5 thousand infected non-isolated agents are present in the same day, the number of negative tests added for each positive would be of 10. This mechanism was implemented so that the optimal testing capacities associated with $R_0 \leq 1$ would always be sufficient to account for a positivity rate below 5%. This threshold was set on May 12[th] 2020 by the World Health Organization as a key requirement to lift or avoid lockdowns [7].

Finally, to determine the optimal testing capacity for the process of contact tracing and testing, we followed a simple heuristic. We initiated the simulations of the fifty scenarios with a value of 0.1 tests for thousand agents, across all conditions. If any of the fifty scenarios resulted in a number of pre-symptomatic and asymptomatic/paucisymptomatic infections above zero by day 60, the capacity was increased by 0.1 for that condition, thereby restarting the process. For the conditions showing that the number of infections kept increasing exponentially, irrespective of the testing capacity, the reported values indicate an approximate threshold that allows the testing capacity not to be exhausted for at least 50 days of simulated time, across all scenarios.

## 2.4 Simulation of testing policies aiding contact tracing

We tested several policies to aid contact tracing and controlled the results in a comparison with either contact tracing and testing alone, or with the same process aided by random sampling in the entire map. The aiding policies consisted in variations of weighted sampling

within a small 'cell', replicating the dimension of the small 'sector' used for the travel behaviour (a square with a side of 100 pixels, cf. Fig 1). To flexibly target the areas in the map reporting the latest outbreaks of new infections, the cells were centred on the coordinates of highest concentration of new infections, as recorded the day prior to the sampling. To determine these coordinates, we defined a cell centred on each new infection recorded the day prior to the sampling and assigned each of them a value equivalent to the number of new known infections (i.e., due to positive test or symptoms onset) included in the cell. The coordinates associated with the highest value would determine the highest concentration of new infections. In case of multiple cells reporting the same number of new infections, a cell would be selected randomly. For instance, the optimal policy found for the condition $NY_2$ consisted in sampling, within a small cell (100x100 pixels) centred on the latest outbreak (i.e., the cell with the highest concentration of new infections), with weights of 60%-20%-20% for the three cohorts of travel behaviour (short, medium, long travel range), whereas the optimal weights found for $seIT_2$ or $Mid_1$ were 20%-40%-40% and 80%-10%-10%, respectively. The potential target of these sampling testing policies were all healthy agents, as well as pre-symptomatic, paucisymptomatic and asymptomatic infections; in other words, all the agents who were not already isolated due to a known infection could be selected for a random test. The number of agents tested in these aiding policies was determined daily, so as to exhaust the testing capacity left unused by the process of contact tracing and testing. The agents found positive would then be isolated (i.e., they could not contribute to the future transmission of the virus) and would be included in the pool of agents to be traced, starting from the subsequent simulated day.

## 2.5 Code specifics and availability

The code was optimised for MATLAB r2019b (MathWorks, Natick, MA), and it allows loading a black and white dot-map of population density to test the effects of the different policies under realistic conditions of population density and geographic distribution. The maps used in the described case studies have been generated relying on data distributed by the Global Human Settlement (GHS) framework (https://ghsl.jrc.ec.europa.eu, CC BY 4.0 license, for the European maps) and the Cooper Center of the University of Virginia (for the New York metropolitan area, https://demographics.virginia.edu/DotMap/index.html). These regions have been chosen only to illustrate that differences in testing policies can emerge when comparing significantly different population distributions and geographical features. In particular, the three maps differ in presenting: 1) a single high-density location with gradually decreasing density as a function of the distance from the centre (New York metropolitan area); 2) a multi-centre organization, with vast areas at very low density or not inhabited (southeast Italy); 3) a completely urbanised region, mostly characterised by uniform low density (Midlands in UK). The script can process any population density dot-map [e.g., see Urban Data Visualisation by Duncan Smith, CASA UCL (https://luminocity3d.org/WorldPopDen)] after these have been converted into grey scale JPEG (e.g., using the free software Gimp, v2.8) and cropped to a format of 1000x1000 pixel, 300dpi, so to have brightest part of the picture representing the highest population density. The resulting images are converted by the script into a matrix of probabilities that matches the grey scale/distribution in the source dot-map: at the beginning of the simulation agents are randomly assigned a position in the map according to these probabilities. The entire codebase for these simulations is freely available as OSF repository (DOI: 10.17605/OSF.IO/QF8WC). Ideally, we hope it can be used as a starting point for more realistic simulations that would allow to develop tailored forms of surveillance testing.

## 3. Results

### 3.1 Estimating the optimal contact tracing capacity

Our first set of simulations covered 60 days across 50 scenarios (i.e., 50 random seeds) and 90 conditions of parameter settings, in a 3 (relative transmission efficiency) x 5 (contact tracing and testing efficacy) x 2 (percentage of asymptomatic or paucisymptomatic infections) x 3 (size of secondary infection cluster) design. Each scenario assumed an initial number of ~50 infected agents, uniformly distributed in a population of 100,000 simulated agents, putatively replicating the prevalence of the disease in the days preceding a lockdown [e.g. see: 37].

   We found that suppression of viral transmission cannot be achieved at low levels of contact tracing, testing, and isolation efficacy, irrespective of testing capacity. Specifically, for sixteen out of the thirty simulated conditions, irrespective of the maximum size of the secondary infection clusters, we found the minimum testing capacity required to find and isolate a sufficient number of infections to suppress viral transmission. In other words, we established the resources required to reach $R_0 < 1$ (Table 1, Fig 4). These conditions were characterised by high ($\geq 60\%$) contact tracing and testing efficacy, and a testing capacity between 0.7 (low transmission efficiency, 20% asymptomatic or paucisymptomatic infections, irrespective of the secondary infection cluster size) and 9.1 (high transmission efficiency, 30% asymptomatic or paucisymptomatic infections, maximum secondary infection cluster size of 25 agents) per thousand agents. Under eight of the remaining 14 conditions (Table 1), the simulations indicated that a testing capacity varying between 0.7 (low transmission efficiency, 20% asymptomatic or paucisymptomatic infections, irrespective of cluster size) and 11.9 (high transmission efficiency, 30% asymptomatic or paucisymptomatic infections, maximum secondary infection cluster size of 25 agents) could contain the virus transmission, but was not sufficient to reduce the number of new daily infections to zero. Instead, the daily number of infected agents remained stable or slightly decreased on average across the simulated scenarios ($R_0 \approx 1$; Fig 4a and 4e). Similarly, for the remaining six conditions characterised by low contact tracing and testing efficacy (20% and 40%) and medium or high transmission efficiency (25% or 35%), the exponential growth of infections could not be contained, at any value of testing capacity ($R_0 > 1$; Table 1, Fig 4c and 4e), irrespective of the maximum size of secondary clustered infections. Interestingly, changes in the parameters controlling the percentage of asymptomatic or paucisymptomatic infections and the secondary infection cluster size affected the resources required for the testing capacity. However, they did not affect the separation into the described three sets of conditions, characterised by $R_0$ below, equal to or above 1 (Table 1). Finally, these results were identically replicated with a population of 1,000,000 agents by randomly sampling only 10% of the scenarios, due to cumbersome computational resources required.

### 3.2 Solutions for the low efficacy contact tracing and testing systemic failure

Our first set of results indicated that, to achieve $R_0 < 0$, it is necessary to either increase the efficacy of contact tracing above 60% (e.g., by enhancing various forms of movement surveillance), or implement robust non-pharmaceutical mitigation strategies, thus reducing the relative transmission efficiency of the virus. As an alternative, surveillance testing (i.e., tests directed towards the general population, irrespective of symptoms) has received increased attention [3, cf. 36]. Our second set of results illustrates the effects of multiple randomised surveillance testing policies in support of contact tracing, relying on a 3 (geographical distribution) x 2 (distributions of travel behaviours) design. Specifically, we simulated the effects of using mixed testing policies (i.e., random-sample or weighted-sample testing in addition to

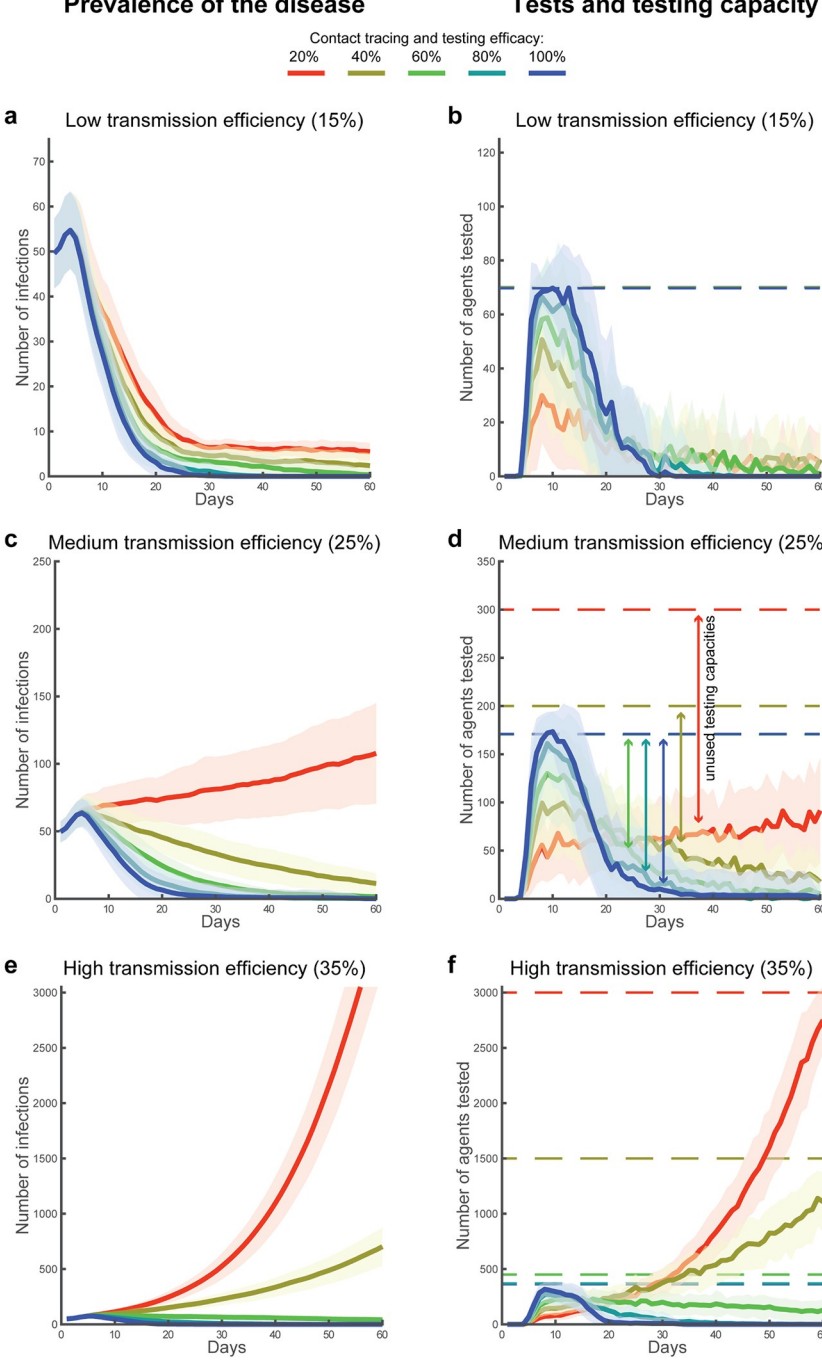

**Fig 4. Disease prevalence, agents tested and capacity.** Error bands (mean and standard deviation) represent the prevalence of COVID-19 in the population over 60 days of simulated time (a, c, e), and the associated number of daily tested agents in relation with the respective testing capacities (solid and dotted lines respectively in b, d, f). The 3x5 design was used to simulate three conditions of simulated transmission efficiency, e.g., due to different mitigation strategies in place, which regulated the growth in the number of infections (a-b: 15%, c-d: 25% and e-f: 35%), and five conditions of contact tracing and testing efficacy (100%, 80%, 60%, 40% and 20%). These simulations illustrate the dynamics found across all conditions of symptomatology in the population distribution and across all conditions of secondary infection cluster size. For these simulations, the percentage of asymptomatic/paucisymptomatic infections is fixed at 20% and the size of the secondary infection cluster is fixed to 1 across all agents (i.e., deterministic setting).

contact tracing and testing) in three geographical maps of population density (New York metropolitan area, Southeast Italy, and the Midlands in UK). These maps were chosen as representative of significantly different population distributions, and they were used in combination with the two described population-wide travel behaviours, with the purpose of illustrating the effects of different ideal sampling strategies in realistic, albeit simplified, settings. For these simulations, we kept constant the relative transmission efficiency (25% daily growth in the number of non-isolated infections), the contact tracing efficacy (20%), the percentage of asymptomatic or paucisymptomatic infections (20%), and the secondary infection cluster size (deterministic setting), as a proof of concept. This second set of simulations covered 60 days, across the same 50 scenarios controlled by the same seeds used for the first set of simulations.

First, the simulations showed that the policy of contact tracing, testing, and isolation mitigated the transmission of the virus in similar ways across all geographical and travel behaviour distributions (Fig 5). The mean number of infected agents recorded at day 60 across the 50 simulated scenarios was 104.74±32.04 and 104.94±31.97, respectively, with the uniform ($NY_1$) and skewed travel distribution ($NY_2$) in the New York metropolitan area. Similarly, for southeast Italy, we found 100.18±31.76 and 101.46±34.81 infected agents in association with the same two distributions of travel behaviours ($seIT_1$ and $seIT_2$, respectively). Finally, for the map of Midlands, UK, we found 109.32±30.36 and 101.4±34.44 infected agents ($Mid_1$ and $Mid_2$, respectively). A 3x2 within-scenario repeated-measures ANOVA revealed no significant effect of the geographical distribution (Sphericity assumed, $F(2,98) = .87$, $p = .42$), distribution of travel behaviours ($F(1,49) = .62$, $p = .43$) or interaction effect ($F(2,98) = .86$, $p = .42$).

Second, we found that the unused capacity available for contact tracing and testing could be employed in further testing policies, marking an improvement in terms of the isolation of infections and thus reducing viral transmission. For instance, contact tracing coupled with random sampling of the entire population significantly reduced the number of infections recorded at the last simulated day, across all scenarios ($NY_1$: 86.62±32.04, $NY_2$: 79.08±31.97, $seIT_1$: 80.6±31.76, $seIT_2$: 88±34.81, $Mid_1$: 86.36±28.83, $Mid_2$: 79.74±28.33 for the 3x2 conditions). A 2x3x2 within-scenarios repeated-measures ANOVA revealed a significant main effect of the testing policy (Sphericity assumed, $F(1,49) = 115.59$, $p < .0001$), but no significant effect of any other factor (geography: $F(2,98) = .19$, $p = .83$; travel behaviour: $F(1,49) = 1.65$, $p = .2$) or interaction of factors (policy*geography: $F(2,98) = 1.04$, $p = .36$; policy*travel behaviour: $F(1,49) = .0$, $p = .97$; geography*travel behaviour: $F(2,98) = 2.58$, $p = .08$; policy*geography*travel behaviour: $F(2,98) = 1.03$, $p = .36$; Fig 5).

Finally, we found that geography- and behaviour-specific policies could further improve the containment of the disease, allowing for containment of viral transmission ($R_0 \leq 1$), despite the low setting of contact tracing and testing efficacy (20%). In particular, we tested different sampling methods where we limited the targeted sampling area to a cell equivalent to the small size sector used for the travel range (Fig 1), and sampled the population giving different probability weights to the different travel cohorts. We found that travel-weighted sampling would successfully aid contact tracing and testing (Fig 5). The optimal weights of these outbreak-centred sampling policies varied as a function of the population distribution on the maps and the distribution of travel behaviours (cf. Fig 5b and 5d). In *t*-test comparisons between these mixed policies and the joint use of contact tracing and random sampling over the entire population, the mixed policies were found to significantly reduce the number of infected agents by day 60 of the simulation, for $NY_1$ (73.3±26.69: $t(49) = 3.29$, $p = .002$; Fig 5a), $NY_2$ (60.34±29.29: $t(49) = 4.12$, $p = .0001$; Fig 5b), $seIT_2$ (69.78±30.51: $t(49) = 4.31$, $p < .0001$; Fig 5d), $Mid_1$ (72.52±27.94: $t(49) = 2.86$, $p < .006$; Fig 5e) and $Mid_2$ (48.68±27.8: $t(49) = 8.49$, $p < .0001$ Fig 5e). Intuitively, where mixed testing policies are in place, an increase in testing capacity results in increased sampling and increased likelihood of finding and isolating pre-

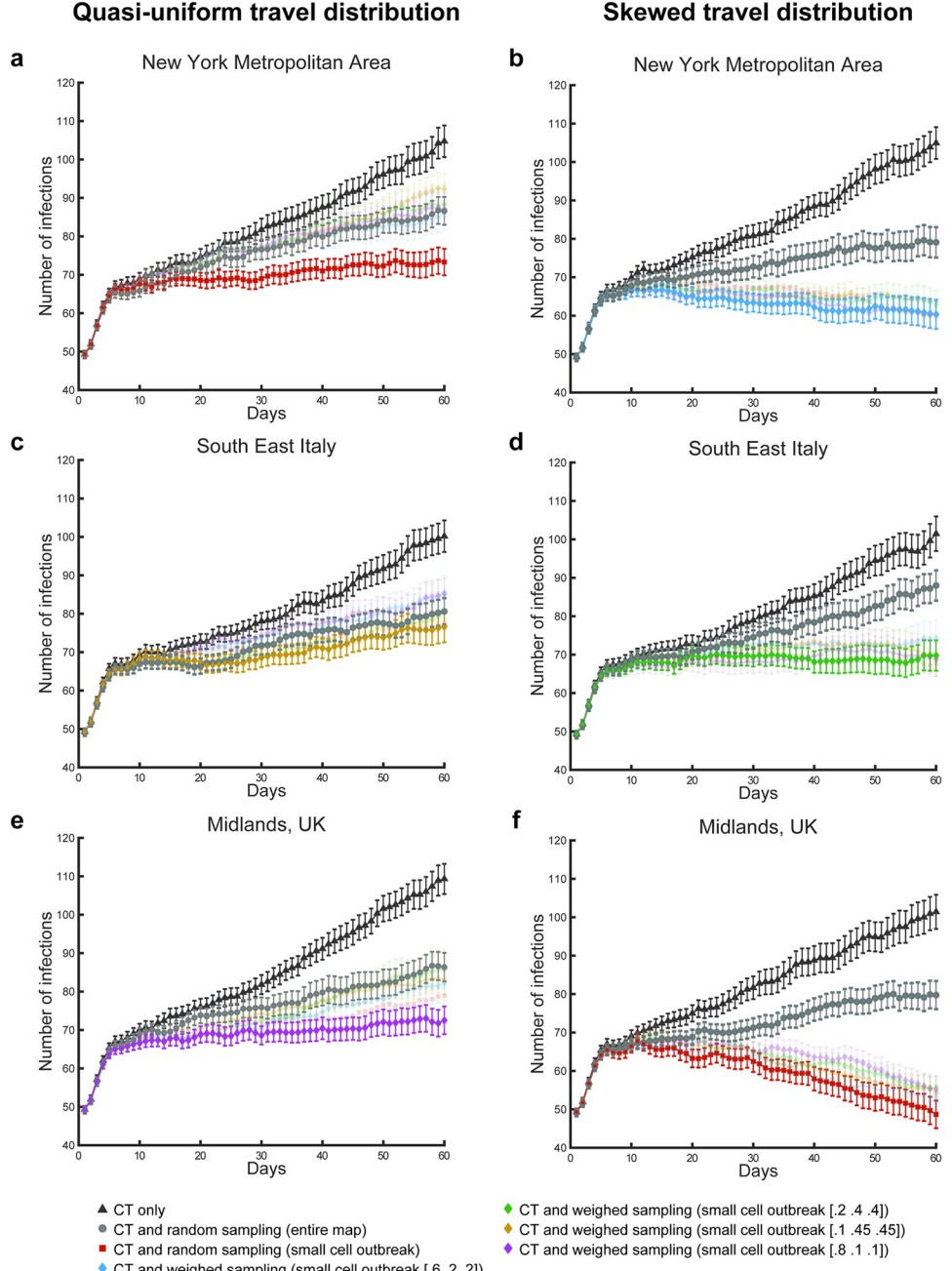

**Fig 5. Effects of mixed testing and isolation policies.** The charts report mean number of infections per day and standard error, associated with different conditions and testing policies. In particular, the panels highlight the different containment results generated using contact tracing and testing (CT), alone (black triangles), contact tracing and testing jointly with random sampling across the entire map (grey circles), and the combination of contact tracing and testing jointly with the best performing sampling policies. Note that these optimal policies change depending on the simulated conditions of geographical distribution and travel behaviour of the population. Under all conditions, the optimal testing policy to aid contact tracing limits the sampling to a cell in the map (100x100 pixel), centred on the coordinates of the most severe outbreak recorded in the previous day of simulated time. For two conditions, the optimal sampling is random within this cell (a, f). For the remaining four conditions (b, c, d, e), different sampling weights are used for the three cohorts of travel behaviour (short, medium, long travel range, respectively).

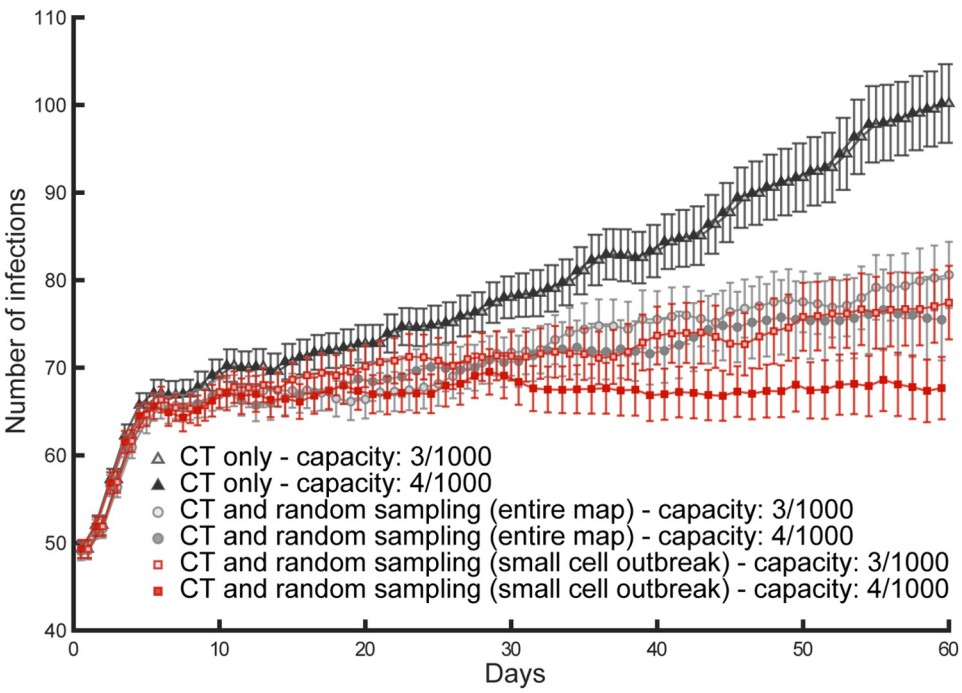

**Fig 6. Increased testing capacity and testing policies.** These simulations illustrate the effects of an increase of testing capacity from 3 to 4 tests per thousand agents, jointly with 20% contact tracing and testing (CT) efficacy, deterministic transmission. The contact tracing and testing process, when considered alone (filled triangles for high capacity and empty triangles for high capacity), does not exhaust the initial testing capacity due to low efficacy, so that an increase in capacity is ineffective as it simply increases the number of unused tests per day. Instead, improved containment of the disease transmission is found both for contact tracing and testing jointly with random sampling over the entire population (filled circles for high capacity and empty circles for low capacity), as well as for contact tracing and testing jointly with random sampling over a small sector centred on the most recent outbreak (filled squares for high capacity and empty squares for low capacity). The latter mixed policy succeeds in keeping the number of daily infections constant ($R_0 \approx 1$), once the capacity is increased.

symptomatic and asymptomatic or paucisymptomatic infections, therefore reducing the $R_0$ of the disease (Fig 6).

## 4. Discussion

The sweeping stay-at-home orders that have been put in place across the world to contain the transmission of the novel coronavirus SARS-CoV-2 were urgently needed to avoid overwhelming the healthcare systems. However, these mitigation strategies have come at a high social and economic cost, leading to massive unemployment, education gaps for students, and social unrest. As multiple countries try to keep industries, services, and educational facilities open, policy-makers are developing strategies that are meant to prevent or contain new waves of infections [6]. Here, we estimated the testing capacity required to identify and isolate a sufficient number of infected subjects to break the chains of transmission of SARS-CoV-2, therefore allowing for a contained impact of the disease, possibly avoiding the most severe mitigation measures, like stay-at-home orders. For these estimations, we used multi agent-based simulations in a soft artificial life approach [33, 34, 49], in place of well-known statistical and mechanistic models [6, 25, 26, 50, 51], so as to highlight emergent properties and dynamics in the interaction between environment, containment policies, and agents' behaviour.

 Our estimations of the testing capacity varied along five key dimensions: 1) the presence of non-pharmaceutical mitigation policies in place (and likelihood of compliance), which affect

the transmission efficiency of the virus and therefore the $R_0$ [24]; 2) the efficacy of contact tracing and testing and the reliability of testing results [27, 28]; 3) the percentage of asymptomatic or paucisymptomatic among the infected [17, 18, 22, 35, 52]; 4) the dimension and frequency of super-spreader events or secondary infection cluster size [48]; 5) the prevalence of the disease in the population [37]. To account for this variability, we used our simulations to explore how the testing capacity would have to change depending on variations over these independent variables, as we tested 3 conditions for the relative viral transmission efficiency, 5 conditions of contact tracing and testing efficacy, 2 conditions for the percentage of asymptomatic or paucisymptomatic infections, 3 conditions of secondary infection cluster size, and 50 scenarios characterised by differences in the disease prevalence at day 1of the simulated time. Finally, these results, acquired in populations of 100,000 agents, were also replicated with reduced sampling (due to the computational resources required) in populations of 1,000,000 agents.

Our simulations provide a few key insights that can guide testing policy development. First, we found that at low levels of contact tracing and testing efficacy, the $R_0$ of the infection remains equal to or above 1, indicating the disease is not suppressed and under some conditions is not even contained, irrespective of the testing capacity. It is important to stress that, for those conditions showing increasing numbers of infections, we set the testing capacity so that it would not be exhausted for at least 50 of the total 60 simulated days. This means that under low levels of contact tracing and testing efficacy, an upsurge of COVID-19 could not be contained with a further increase in daily capacity (Fig 4c–4f). These results suggest that low efficacy leads to a systemic failure in disease containment which is independent of the availability of tests *per se*. Instead, in our simulation, the high percentage of missed contacts enhanced a predator-prey dynamic (i.e., Lotka-Volterra non-linearity [53]; Fig 1), where the predators (the tests) lost track their prey (the infections), and remained idle (part of the daily test capacity remained unused). This predator-prey dynamic allowed the disease to spread undetected, leading to a minimally-mitigated wave of infections. This is a striking result considering that most countries indeed failed to uncover the real dimensions of the disease prevalence, as clearly demonstrated in recent seroprevalence studies [30, 31, 37], and still indicated at present by the significant difference between case fatality rates [2] and the estimated mortality rate [37, 38]. For instance, the case fatality rate in the US, limited to the cases found positive in summer, rests at about 2% [54], against an estimated mortality rate of about 0.5%-1% [37, 38], suggesting that only between 25% and 50% of the total infections are detected. This result may also explain why countries prepared with a diffused network for contact tracing and a high testing capacity in the early months of the pandemic were not able to use their available capacity and failed to contain COVID-19, eventually resorting to stay-at-home orders (e.g., as hypothesised in the case of Germany [55]). Our simulations indicate that improvements in contact tracing and testing efficacy, e.g., due to increased reliability of test results or increasing mass tracing capabilities, are required to exceed the 60% threshold. Beyond this value, jointly with the safe isolation of all symptomatic subjects and those who tested positive, we found a steady decline in the number of infections across all simulated conditions of transmission efficiency, as well as across conditions of asymptomatic/paucisymptomatic infections, or size of secondary infection cluster. This result is consistent with a recent mechanistic model estimation [23], demonstrating robustness of the finding across theoretical constructs. The viral transmission for SARS-CoV-2 seems to be intrinsically associated with reduced contact tracing efficacy, possibly due to the relevance of airborne transmission [56, 57]. Therefore, significant improvements in the efficacy of contact tracing and testing, aimed at reaching the 60% threshold, may only come at the expenses of a vast expansion of mass surveillance efforts.

As an alternative to this route, which poses legal and ethical problems for democratic societies, governmental agencies like the Center for Disease Control and Prevention (CDC) in the United States and the European Centre for Disease Prevention and Control (ECDC) for the European Union have released new guidance to encourage surveillance testing, or in other words campaigns of tests directed at the general population irrespective of symptoms [3, 36]. We expanded on this idea in our second set of simulations, under the assumption that unbiased random testing in aid of contact tracing and testing might not always be the optimal strategy. This set of simulations is meant to illustrate that general information about population distribution and range of movement can ideally be used to improve the containment of the disease. We used multiple sampling policies to aid the monitoring of new outbreaks and feed missed contacts to the main process of contact tracing and testing. We found that, while the process of contact tracing and testing is agnostic to both the geographic distribution and the population-level behaviours, optimal aiding policies are shaped by the features of the environment and cohort-level behaviour [cf. 58, 59]. Although the mixed testing policies we tested in our second set of simulations were developed only for illustrative purposes, they show that uniform random sampling of the population might not always be the optimal strategy to improve containment. Population-level information can instead be used to guide test surveillance strategies, reducing the $R_0$ below 1, even under conditions of low efficacy of contact tracing and testing.

The current study has a few limitations due to the simplifications that have been incorporated into the simulations. Aside from considerations concerning the unknown characteristics of viral transmission that are still to be determined for SARS-CoV-2, and whose impacts are difficult to estimate, important simplification concern: 1) the choice of the initial prevalence; 2) the assumption of a "closed" environment (i.e., no infections are added during the simulated time as coming from somewhere outside of the maps); and 3) the assumption that all symptomatic subjects are isolated. For the first two points, intuitively, the estimated testing capacity would have to be increased if the prevalence at the beginning of the simulated time had to increase or a constant inflow of new infections were to be included in the scenarios. Because the objective of the study was to aid policymakers in preventing new uncontrolled surges or waves of infections following a period of lockdown, we chose to test for 50 undetected infections per 100,000 agents, at day 1. For instance, the number of infections reported in late July for the regions used as case studies (i.e. after these regions lifted their respective lock-down measures) was well below the prevalence used for our simulations, even when considering that the majority of the infections were undetected. Less than one thousand infections have been reported on average in July for the entirety of Italy, UK, and for the combined states of New York and New Jersey. Conversely, the areas used as case study would require respectively ~2, ~4 and ~10 thousand infections to match the prevalence setting used for our simulations.

Concerning the third simplification and the immediate isolation of symptomatic infections, we acknowledge that this assumption implies that agents displaying symptoms would have immediate access to a test and would be able to either self-isolate or be isolated in a healthcare facility, thus avoiding even household transmission. While this choice allowed us to avoid further clinical assumptions (e.g., relative percentage of severe and mild infections, or time prior to hospitalization, when necessary), as well as environment specific settings (e.g., access to sick leave, willingness to comply with social norms, political ideology, etc.), it focussed our investigation on the pre-symptomatic and asymptomatic/paucisymptomatic transmission, which plays a fundamental role in the COVID-19 pandemic [17, 18, 22]. Furthermore, since the cohort of the asymptomatic/paucisymptomatic infections is only isolated after a positive test, we can infer that a less stringent isolation rule for the symptomatic infections would lead to similar results as those reported in association with an increase in the percentage of the non-

isolating cohort (from 20% to 30%, cf. Table 1). Therefore, we would expect to find an increase in terms of the contact tracing and testing resources, when containment is possible, and exponential growth under conditions of low contact tracing and testing efficacy, irrespective of the testing capacity.

In conclusion, the described systemic failure under conditions of low contact tracing and testing efficacy was found across a wide range of parameters, including variations in the percentage of asymptomatic or paucisymptomatic infections, population size, population distribution, relative transmission efficiency and secondary infection cluster size. This information can help policymakers plan for the resources required by contact tracing and testing in a given region, while at the same time developing context-specific test surveillance policies, tailored for the specific characteristics of population density, distribution, and population-level behaviour.

## Author Contributions

**Conceptualization:** Vincenzo G. Fiore, Nicholas DeFelice, Benjamin S. Glicksberg, M. Andrea Pisauro, Xiaosi Gu.

**Formal analysis:** Vincenzo G. Fiore, Ofer Perl, Anastasia Shuster.

**Investigation:** Vincenzo G. Fiore, Nicholas DeFelice, Benjamin S. Glicksberg, M. Andrea Pisauro, Xiaosi Gu.

**Methodology:** Vincenzo G. Fiore, Dongil Chung, Xiaosi Gu.

**Resources:** Vincenzo G. Fiore, Ofer Perl, Anastasia Shuster, Kaustubh Kulkarni, Madeline O'Brien, M. Andrea Pisauro, Xiaosi Gu.

**Software:** Vincenzo G. Fiore.

**Supervision:** Dongil Chung, Xiaosi Gu.

**Visualization:** Ofer Perl, Anastasia Shuster, Kaustubh Kulkarni, Madeline O'Brien.

**Writing – original draft:** Vincenzo G. Fiore, Nicholas DeFelice, Benjamin S. Glicksberg, Dongil Chung.

**Writing – review & editing:** Vincenzo G. Fiore, Nicholas DeFelice, Benjamin S. Glicksberg, Ofer Perl, Anastasia Shuster, Kaustubh Kulkarni, Madeline O'Brien, M. Andrea Pisauro, Dongil Chung, Xiaosi Gu.

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
