## [Decision Letter · Decision Letter 0]

14 Oct 2020

PONE-D-20-24957

Containment of COVID-19: simulating the impact of different policies and testing capacities for contact tracing, testing, and isolation

PLOS ONE

Dear Dr. Fiore,

Thank you for submitting your manuscript to PLOS ONE. After careful consideration, we feel that it has merit but does not fully meet PLOS ONE’s publication criteria as it currently stands. Therefore, we invite you to submit a revised version of the manuscript that addresses the points raised during the review process.

Please address all the comments of the reviewers and the AE. In particular, I don't think the so-called semi-static model is suitable for agent-based modeling. Please make the model stochastic and controlled by widely accepted parameters such as transmission rate or basic reproductive number. 

We look forward to receiving your revised manuscript.

Kind regards,

Yang Yang, Ph.D

Academic Editor

PLOS ONE

Additional Editor Comments:

In this manuscript, the authors presented some modeling results about how testing capacity, contact tracing & testing efficacy, disease incidence and proportion of asymptomatic infections jointly determine the spread of COVID-19. The most interesting finding is that, regardless of testing capacity, a low efficacy would lead to continuing growth of the epidemic.

My major concern with this manuscript is the deterministic way for generating transmission dynamics. This is the first time I saw a deterministic (or semi-static) agent-level modeling, although population-level deterministic models have been used widely. The stochasticity of secondary transmissions is the major advantage of agent-based modeling that makes it a better approximation of the real-world epidemics. I don’t think sacrificing this advantage for computational efficiency is justifiable. The number of negative tests were also generated deterministically, with which I’m OK. The magnitude of each epidemic was calibrated to yield desired growth in disease incidence rather than controlled by the more widely accepted threshold parameters such as the basic or effective reproduction number, which also looks questionable to me. We need to know what are the corresponding R0 for the different disease incidence settings, but I don’t see a clear way to establish the link. Even if such a link can be established, we need to be convinced that the corresponding R0 has the threshold property.

Specific comments:

1. Page 4, line 83, what shortcomings?

2. Page 5, lines 119-120, “environmental conditions” is a misleading phrase which is often interpreted as, e.g., climatic conditions. Maybe “parameter settings”?

3. Page 5, lines 130-131. The contact tracing & testing efficacy sounds like “sensitivity” in a broad sense, i.e., combining the success rate of contact tracing and sensitivity of the biological test. You may want to articulate more about its interpretation. I’m confused by “any agent who tested positive” in line 131. Why do you need the infectors to be tested positive? Don’t you assume infected people who are tested negative falsely remain infectious and can transmit?

4. Page 6, lines 166-169. The numbers set for the conditions with growing numbers of infections (i.e., the numbers in dark gray cells of Table 1) are not straightforward to interpret, even after referring to the methods (lines 451-458). Are these numbers practical meaningful or informative?

5. Where is Figure 4?

6. I’m not sure how meaningful it is to simply embed 100,000 individuals in the three real regions (New York, Southeast Italy and Midlands). The obtained epidemics will almost surely not resemble what truly occurred in these regions. Again, an epidemiologically meaningful way is to calibrate a stochastic or deterministic transmission model to the real disease incidence data in these regions to estimate key parameters such as transmission rate or R0, and then simulate epidemics under hypothetical contact tracing and testing conditions using these key parameters.

7. Page13-14, lines 399-400, as all cases were assumed isolated upon symptom onset, is it meaningful to simulate recovery time or ICU time?

8. Page 14, lines 414-428, once an infectious person was chosen to generate a transmission, will this person be eligible again for future selection of transmitters?

9. Page 15, line 432, “These were randomly selected”, how is the amount of such selection of cases for contact tracing determined? Does it depend on assumed testing capacity?

10. Page 15, line 449, The ratio of untested infected over healthy agents should be most below 1, how can it be equal to the number of negative tests? Why is it necessary to set a maximum (20) for the number of negative tests per each positive one? Is this maximum number supported by any reference?

11. Page 16, lines 465-467, how is ‘concentration” calculated? How do you define “latest outbreak”?

12. Figure 3, many colored curves cannot be seen clearly. Can you use smaller symptoms and thinner lines? If some settings do not really differ much from each other, you may consider dropping some settings.

Journal Requirements:

2) We note that you have stated that you will provide repository information for your data at acceptance. Should your manuscript be accepted for publication, we will hold it until you provide the relevant accession numbers or DOIs necessary to access your data. If you wish to make changes to your Data Availability statement, please describe these changes in your cover letter and we will update your Data Availability statement to reflect the information you provide.

3) Please upload a copy of Figure 4, to which you refer in your text. If the figure is no longer to be included as part of the submission please remove all reference to it within the text.

4) We note that Figure 4 in your submission contain map/satellite images which may be copyrighted. All PLOS content is published under the Creative Commons Attribution License (CC BY 4.0), which means that the manuscript, images, and Supporting Information files will be freely available online, and any third party is permitted to access, download, copy, distribute, and use these materials in any way, even commercially, with proper attribution. For these reasons, we cannot publish previously copyrighted maps or satellite images created using proprietary data, such as Google software (Google Maps, Street View, and Earth). For more information, see our copyright guidelines: http://journals.plos.org/plosone/s/licenses-and-copyright.

i.    You may seek permission from the original copyright holder of Figure(s) [#] to publish the content specifically under the CC BY 4.0 license.

ii.    If you are unable to obtain permission from the original copyright holder to publish these figures under the CC BY 4.0 license or if the copyright holder’s requirements are incompatible with the CC BY 4.0 license, please either i) remove the figure or ii) supply a replacement figure that complies with the CC BY 4.0 license. Please check copyright information on all replacement figures and update the figure caption with source information. If applicable, please specify in the figure caption text when a figure is similar but not identical to the original image and is therefore for illustrative purposes only.

Reviewers' comments:

Reviewer's Responses to Questions

**Comments to the Author**

1. Is the manuscript technically sound, and do the data support the conclusions?

Reviewer #1: Yes

Reviewer #2: Yes

2. Has the statistical analysis been performed appropriately and rigorously? 

Reviewer #1: I Don't Know

Reviewer #2: Yes

3. Have the authors made all data underlying the findings in their manuscript fully available?

Reviewer #1: Yes

Reviewer #2: Yes

4. Is the manuscript presented in an intelligible fashion and written in standard English?

Reviewer #1: Yes

Reviewer #2: Yes

5. Review Comments to the Author

Reviewer #1: Using simulation models, this article demonstrates the importance of contact tracing and testing (especially the former) in controlling the COVID-19 epidemic, and I recommend that it should be published with minor revisions. The manuscript appears technically sound, and the data support the conclusions.

I suggest the overall format of the article be revised, as the results are currently presented before the methods. It would be more conventional and easier to understand if the methods are presented first. Some figures that are referred to in the text were not included in the manuscript, which was a bit confusing.

I have a few suggestions for minor revisions:

1. Line 89-115: these 2 paragraphs don’t really fit into the introduction, as they include a mixture of research methods and results. It would be easier to read the article if all the methods are first presented separately (in a methods section), and then the results presented. Eg. Line 92-102 and 108-111 describe methods; 103-104 and 112-115 describe results. All the “see methods” comments in the results section can be removed if the results are presented after the methods.

2. Line 100 should read “due to false negatives”

Line 307 should read “irrespective of symptoms”

Line 397-400: (The number of days required to develop symptoms, if any (Figure 4b), the number of days to reach full recovery, with a bimodal distribution due to the shorter time of recovery for the asymptomatic (Figure 4c), and the time spent in intensive care units (Figure 4d) were also predetermined). I suggest these technical details be expanded and clarified: what exactly were the predetermined numbers that were used in the models for the incubation period, recovery period and intensive care unit period? Also, the figures referred to were not included in the manuscript, so I cannot comment on them (Figure 4a-d are missing from the manuscript).

Reviewer #2: This manuscript discusses potential methods to contain the COVID-19 through the multi-agent simulations. This research is really interesting, and the authors explained their goals, simulation methods, and the results comprehensively. Here are my comments:

1) The whole manuscript needs review and edition in terms of English proficiency. Examples are lines 46-53.

2) Lines 55-58 explain some suggestions made by WHO, but there is no reference to the WHO link for these suggestions.

3) In lines 89-90, the authors declared that they focused on pre-symptomatic and asymptomatic cases. They also assumed symptomatic patients would not transfer the disease. This assumption is a little strict and may not be satisfied in reality. Some symptomatic patients may still transmit the disease unless they confirmed positive. It is suitable if the authors explain why they removed symptomatic cases from possible transmission sources or run the simulation by considering this group of patients as a potential infection source.

4) Why two factors, contact tracing, and testing efficacy, are considered together? Especially, in lines 440-441, how they calculated 20% of contact tracing and testing efficacy might result from 25% of tracing with 80% test reliability. It would be better if they could provide more details about this strategy.

5) Lines 136-137: "... would not show symptoms severe enough to induce... . Therefore in our simulations, these asymptomatic agents ... ". A symptomatic ( no matter mild or severe )patient may not be included in the asymptomatic category. Suppose the authors want to have these types of patients in the asymptomatic category. In that case, it is suitable they define their asymptomatic group and explain that this group contains both patients with no symptoms or those who do not show symptoms severe enough to induce self-isolation or hospitalization.

6) line 145: Define effective reproduction.

7) Review lines 172-176.

8 ) In Figure 1, Is the contact tracing efficacy means contact tracing and testing efficacy? If yes, please correct it. If not, where the authors include testing efficacy in this figure?

9) Lines 279-281: The authors claim in low contact tracing and testing efficacy, the number of daily infections keeps growing regardless of the testing capacity. In Figure 1, the number of daily infections does not have increasing trend for low incidence even in low levels of contact tracing and testing efficacy( If we assume the authors means contact tracing and testing efficacy instead of contact tracing efficacy). Moreover, 40% of contact tracing and testing efficacy is enough to contain a disease with medium incidence.

10 ) In lines 293-296: The authors explain that "beyond the 60% threshold of contact tracing and testing efficacy, there is a steady decline in the numbers of infection across all simulated levels of incidence and across all levels of asymptomatic infections." Still, there is no figure or table to support this explanation for the asymptomatic level of 30%. Figure 1 shows the results only for the asymptomatic level of 20%.

11) Lines 351: Please explain what "we simulated the reproduction of the virus in terms of ..." means.

12) Please add Figure 4 to the pdf file

13) Line 402: Please explain the "range of virus transmission."

14) Line 423: Why the authors supposed each contact will result in infection with probability one when the contract is between an infector and a susceptible person? Some contacts may not yield an infection.

15) Please explain why the null propagation of infection (Line 425) simulates a diminished effective reproduction(number)?

16) Line 675: Typo: counties - countries

17) Figure 2 needs more explanation

18) The reference style is not consistent for all the references.

Overall, this manuscript presents an interesting idea and the results validate the idea.

6. PLOS authors have the option to publish the peer review history of their article (what does this mean?). If published, this will include your full peer review and any attached files.

Reviewer #1: No

Reviewer #2: No

---

## [Author Response · Author response to Decision Letter 0]

10 Dec 2020

Dear Dr. Yang,

Thank you, the additional editor and the reviewers for these constructive comments. We have now made every effort to address all major and minor concerns. 

In particular, we have run a new series of agent-based simulations to include a stochastic form of transmission, as requested by the additional editor. This new simulation allows a transmitter to reach up to 25 agents in a single simulated day, thus creating the conditions for the simulation of the so-called super-spreaders. This stochasticity in the transmission of the disease is also used to replicate the recently reported data suggesting 80% of secondary infections originate from 10% of infected people (Endo et al., 2020). 

The new version of the manuscript now also includes a description of how the chosen variables regulating the disease incidence relate with the reproduction number (R0), showing the settings are consistent with known estimates reported in literature about COVID-19. The manuscript has also been restructured as suggested by one of the reviewers, with the results following the methods section. 

Finally, we revised the figures to exclude potentially copyrighted material. The amended figure 1 (previously figure 2) now includes only images that are based on a manipulation of data licensed as CC 4.0 by the European Commission (Global Human Settlement framework).

We hope you will find the revised manuscript appropriate for publication in your journal.

Please see our point-by-point response to the reviewers in the uploaded rebuttal file.

Sincerely,

Vincenzo Fiore, PhD

Icahn School of Medicine at Mount Sinai – Department of Psychiatry 

1 Gustave L. Levy Place, New York, NY 10029-5674

---

## [Decision Letter · Decision Letter 1]

18 Jan 2021

PONE-D-20-24957R1

Containment of COVID-19: simulating the impact of different policies and testing capacities for contact tracing, testing, and isolation

PLOS ONE

Dear Dr. Fiore,

Thank you for submitting your manuscript to PLOS ONE. After careful consideration, we feel that it has merit but does not fully meet PLOS ONE’s publication criteria as it currently stands. Therefore, we invite you to submit a revised version of the manuscript that addresses the points raised during the review process.

We look forward to receiving your revised manuscript.

Kind regards,

Yang Yang, Ph.D

Academic Editor

PLOS ONE

Additional Editor Comments (if provided):

The authors have indeed improved this manuscript. I appreciate their efforts in adding a substantial amount of simulation results. Particularly, I’m glad to see the stochastic transmission being added. The addition of super spreader events is definitely a plus. Nevertheless, after reading the revised manuscript, I still found a lot of questions about the simulation procedure to be clarified, especially the contact tracing part.

1. Lines 107-108, “transmit the virus to up to 1, 8 or 25 healthy agents”, is it on each day or during the whole infectious period? My understanding from the methods is the former.

2. Line 135, “we assume all symptomatic infections entered isolation the day they displayed symptoms.” Do you assume this for infected agents who were missed by contact tracing and testing? If so, make it clear in the discussion as a limitation that household transmission of non-hospitalized symptomatic infections is ignored.

3. Line 145, what is “divided by four” for subclinical cases, mean of duration or the actual sampled duration? Why use Pearson distribution? There are different types of Pearson distributions. Which one is talked about here?

4. Line 146, is it necessary to simulate time in ICU? I don’t see how it impacts the results, as all symptomatic agents have already been isolated since symptom onset.

5. The name “disease incidence” (15%, 25% and 35%) is very misleading. My understanding is these are percentages of infected agents that can actually transmit the virus, which is less than 1 because of nonpharmaceutical interventions other than contact tracing, isolation and quarantine. Why don’t you call it “relative transmission efficiency” or something like that?

6. Lines 179-181, Rt also depends on the number of infections each infected agent can generate per day, or equivalently, the so called “size of infection cluster”. There are several misnomers or unnecessary new terms in this manuscript. “Size of secondary infections” would sound much better than “size of infection cluster”.

7. Line 185, I don’t see how the R0 (not Rt) values (3.05, 2.18 and 1.31) were calculated based on 35%, 25% and 15%. I guess you integrated over the Pearson distributions of the infectious period. Please give the exact formula. Also make it clear this is R0 of only a symptomatic case or a randomly selected infected agent (which could be subclinical).

8. A lot of details about how contact tracing is simulated is still missing. Line 224, “a day-to-day pool of all agents displaying symptoms or found positive…”. What does “found positive” mean exactly? If by testing, when was the testing done? Per my understanding, your simulator does not distinguish between “contact tracing” and “tested positive”. I assume every active symptomatic agent, if not yet “tested and traced”, is eligible for testing and contact tracing. Whether this person will be tested and traced is random with the tracing and testing efficacy parameter as the probability.

9. Line 226 “randomly selected one by one…”. What does “randomly selected” mean here, with what probability? Are all agents in this pool to be tested and traced? This comes back to the question of who were included in the pool. If people in this pool have not been tested and traced yet, then random selection makes for testing and tracing sense.

10. Line 228, “agents (if any) that she had infected at any time…”. Do you mean the agents she or he infected both before and after the current time will be traced? On the current time step, how did you set agents that will be infected later to be traced?

11. Will asymptomatic/paucisymptomatic be tested and traced? If not, why? If yes, when did isolation start?

12. What does “tracing and testing” actually do to impact the simulation? Are traced and tested agents isolated immediately, regardless of symptom display? You mentioned all symptomatic agents will self-isolate upon symptom onset. So the only difference contact tracing makes is to isolate infected contacts of identified agents (who have been tested positive and chose for tracing further contacts) before symptom onset, is my understanding right?

13. Lines 236-237, “but it would also leave undetected the entire chain of transmission associated with that missed agent”. I’m really confused here. Do you mean all contacts of the missed agent, and contacts of the contacts, will all be missed? If so, how do you make the pool mentioned in lines 224-225? On the other hand, for agents tested positive and chosen for contact tracing, how far will the contact tracing go, i.e., do you stop after his or her infected contacts were identified and isolated, or you move on to trace contacts of his or her infected contacts?

14. Lines 245-246, please give explicit calculations to show the equivalence, e.g., (1-25%) + 25% * (1-80%)=80%.

15. Lines 256-257, these 5000 infected and non-isolated agents are not necessarily going to be traced and tested on that day; therefore, isn’t adding 10-20 negative tests per active infected agent too many?

16. Regarding testing policies aiding contact tracing described in lines 273-292, do you sample any agent, regardless of infection status and symptom status, or only sample infected but non-isolated agents (before symptom onset and not traced yet)? If you mean the latter, it seems infeasible in reality. If the former, the underlying population seems too big, and it’s not clear how sampling is done. Do you use all the remaining test capacity for this sampling?

17. In line 376, you mentioned “mean number of infected agents recorded AT day 60”. In line 389, you stated “the number of infections recorded BY the last simulated day”. Which is correct?

Reviewers' comments:

Reviewer's Responses to Questions

**Comments to the Author**

1. If the authors have adequately addressed your comments raised in a previous round of review and you feel that this manuscript is now acceptable for publication, you may indicate that here to bypass the “Comments to the Author” section, enter your conflict of interest statement in the “Confidential to Editor” section, and submit your "Accept" recommendation.

Reviewer #1: All comments have been addressed

Reviewer #2: All comments have been addressed

2. Is the manuscript technically sound, and do the data support the conclusions?

Reviewer #1: Yes

Reviewer #2: Yes

3. Has the statistical analysis been performed appropriately and rigorously? 

Reviewer #1: Yes

Reviewer #2: Yes

4. Have the authors made all data underlying the findings in their manuscript fully available?

Reviewer #1: Yes

Reviewer #2: Yes

5. Is the manuscript presented in an intelligible fashion and written in standard English?

Reviewer #1: Yes

Reviewer #2: Yes

6. Review Comments to the Author

Reviewer #1: (No Response)

Reviewer #2: Thank you so much for addressing my previous concerns. I believe the current version is acceptable for publication after minor revisions of the following comments.

1) Lines 344-345: Please correct the order of the numbers related to medium or high incidence to be compatible with the text.

2) Lines 450-451: Rewrite "we set the testing capacity so that it would not be 450exhausted during most of the simulated time(i.e., more than 50 simulated days".

7. PLOS authors have the option to publish the peer review history of their article (what does this mean?). If published, this will include your full peer review and any attached files.

Reviewer #1: No

Reviewer #2: No

---

## [Author Response · Author response to Decision Letter 1]

22 Jan 2021

Please see our point-by-point response to the reviewers in the attached rebuttal file.

---

## [Editor Report · Decision Letter 2]

10 Feb 2021

Containment of COVID-19: simulating the impact of different policies and testing capacities for contact tracing, testing, and isolation

PONE-D-20-24957R2

Dear Dr. Fiore,

We’re pleased to inform you that your manuscript has been judged scientifically suitable for publication and will be formally accepted for publication once it meets all outstanding technical requirements.

Kind regards,

Yang Yang, Ph.D

Academic Editor

PLOS ONE
---

## [Editor Report · Acceptance letter]

22 Mar 2021

PONE-D-20-24957R2 

Containment of COVID-19: simulating the impact of different policies and testing capacities for contact tracing, testing, and isolation 

Dear Dr. Fiore:

I'm pleased to inform you that your manuscript has been deemed suitable for publication in PLOS ONE. Congratulations! Your manuscript is now with our production department. 

Kind regards, 

on behalf of

Dr. Yang Yang 

Academic Editor

PLOS ONE